# Exploring Influential Factors Shaping Nursing as a Profession and Science in Healthcare System—A Systematic Literature Review

**DOI:** 10.3390/healthcare13060668

**Published:** 2025-03-19

**Authors:** Vlora Ejupi, Allison Squires, Brigita Skela-Savič

**Affiliations:** 1Angela Boškin Faculty of Health Care, Spodnji Plavž 3, 4270 Jesenice, Slovenia; 2Global Consortium of Nursing and Midwifery Studies & Rory Meyers College of Nursing, New York University, New York, NY 10010, USA; aps6@nyu.edu

**Keywords:** nursing professionalism, communication skills, resilience, nursing education, cultural perceptions

## Abstract

**Introduction**: Nursing faces numerous challenges amidst broader socio-political transitions in many countries. Despite efforts to establish formal nursing education and legislative frameworks, the profession’s status remains relatively low within some healthcare systems. This study aims to examine the factors influencing nursing professionalism and provide insights into strategies for its enhancement. **Aim:** The study seeks to summarize the existing literature on nursing professionalism, assess methodological quality, and derive recommendations for future research. Through a systematic review, the study explores various factors shaping nursing professionalism, including communication skills, resilience, education, and cultural perceptions. **Methods:** A systematic search was conducted across multiple electronic databases from 2014 to 2024 to identify relevant studies on nursing professionalism. The search strategy encompassed elements of professionalism, nursing, and exclusion criteria. Selected studies underwent methodological quality assessment using the Critical Appraisal Skills Programme Qualitative Checklist. **Results:** The review identified 421 records, yielding 72 eligible studies after screening and eliminating duplicates. Fourteen studies met the inclusion criteria, revealing a complex framework of factors influencing nursing professionalism. The key determinants include communication skills, resilience, education, and cultural perceptions. Studies emphasized the importance of effective communication, resilience in crisis management, quality education, and understanding psychological and cultural influences on professionalism. **Conclusions:** Nursing professionalism is influenced by multifaceted factors, including communication skills, resilience, education, and cultural perceptions. Recognizing these factors is vital for promoting high-quality nursing practice and ensuring patient safety. Addressing these elements can inform targeted interventions to enhance nursing professionalism and support nurses in delivering optimal healthcare. This study underscores the importance of investing in nursing education, fostering resilience, and considering cultural nuances to cultivate a culture of excellence and professionalism within healthcare systems.

## 1. Introduction

This study conducts a systematic review to examine the factors influencing nursing professionalism, providing insights into strategies for its enhancement.

Nursing is a profession that faces numerous challenges worldwide, reflecting the complex and dynamic nature of healthcare systems across different countries and regions [1]. For instance, in countries like the United States, India, and Brazil, these challenges mirror the broader political, legal, and socio-economic landscapes in which healthcare operates, and they often impact the status, recognition, and development of nursing as a profession and a scientific discipline [2,3].

Formal nursing education has been established in various countries over the years, with different approaches to curriculum development, accreditation, and regulation [4]. While some countries have well-established systems for nursing education and practice, others, such as Kosovo, may still be in the process of developing and formalizing their nursing education programs [5]. In Kosovo, the education system is evolving, with efforts to standardize nursing curricula amid challenges related to limited resources and infrastructure. The health system is public but faces significant issues, including underfunding and a shortage of qualified healthcare professionals. Moreover, while there are laws regulating the nursing profession, the implementation and standardization of these regulations remain inconsistent. Kosovo has historically had a functioning education and healthcare system. These sectors have faced significant disruptions in recent years, leading to inadequate infrastructure, underfunding, and inconsistent implementation of existing laws.

One of the key challenges faced by nursing globally is the variability in standards of practice and education [6]. For example, in Nigeria and Indonesia, there may be a lack of standardized nursing practice and educational protocols, leading to inconsistencies in the quality of care provided by nurses [7,8]. Additionally, there may be limited opportunities for nurses to engage in clinical or leadership roles, develop independent care plans, or conduct scientific research based on critical thinking and contemporary trends [9]. These factors can contribute to a perception of nursing as a less prestigious profession compared to other healthcare professions, and they may hinder the recruitment and retention of talented individuals within the field.

Legislation governing the healthcare sector is another area of concern for nursing professionals worldwide [10]. In countries like India and South Africa, while there are laws and regulations that define the position of nurses in both professional and educational contexts, there may still be inconsistencies or gaps in these laws that impact the practice and recognition of nursing. For example, some countries may lack clear licensure requirements for nurses, leading to confusion and uncertainty among both nurses and employers [11]. Additionally, there may be challenges in ensuring that nursing education programs meet accreditation standards and prepare students adequately for the demands of modern healthcare practice [12].

Efforts to advance nursing often require collaboration between government agencies, professional associations, educational institutions, and other stakeholders [13]. In countries like Canada and Germany, strategic planning initiatives are underway to develop comprehensive strategies for nursing development, encompassing education, practice, regulation, and research [14,15]. These initiatives may involve the input of international experts and the adaptation of the best practices from other countries to suit local contexts. However, the implementation of these strategies may be hindered by resource constraints, competing priorities, and resistance to change within healthcare systems [16].

In some countries, the establishment of professional bodies, such as nursing associations or chambers of healthcare professionals, has helped to enhance the autonomy and organization of the nursing profession [17]. These bodies may be responsible for regulating membership, setting competency standards, overseeing disciplinary measures, and advocating for the interests of nurses and their patients [18]. By providing a unified voice for the profession, these organizations can help to shape healthcare policy and practice in ways that support nursing excellence and promote the delivery of high-quality patient care [19]. 

In our study, we aim to approach the concept of nursing professionalism primarily from a meso-level institutional perspective, focusing on the internal and organizational factors that shape professional conduct within healthcare settings. While we recognize the importance of macro-level influences, such as government policies and national health regulations, our emphasis is on how institutional culture, leadership, professional development opportunities, and peer interactions influence nursing professionalism. This aligns with the work of [20], who also explore the multifaceted nature of nursing professionalism, emphasizing the complexity of factors—ranging from individual motivations to broader institutional frameworks—that shape professional behavior. Our approach seeks to further differentiate these internal factors, allowing us to understand how organizational dynamics directly impact the professional growth and conduct of nurses.

Despite these institutional efforts, nursing may still face challenges in achieving desired standards within healthcare systems. National health strategies and policies may lack specific objectives or action plans aimed at advancing nursing care, highlighting the need for continued advocacy and engagement by nursing stakeholders [21,22]. Additionally, in countries such as Japan and Mexico, there may be cultural or societal barriers that contribute to the undervaluation of nursing as a profession, such as gender biases or stereotypes about the nature of nursing work [23]. Addressing these barriers requires a concerted effort from all stakeholders to promote the value of nursing and ensure that nurses have the resources, support, and recognition they need to thrive in their roles [24]. 

While progress has been made in formalizing nursing education and institutionalizing the profession in many countries, significant challenges persist in elevating nursing’s status within healthcare systems worldwide [25]. Moving forward, sustained efforts are needed to promote the development of nursing as a science and ensure its integration into evolving healthcare landscapes globally [26]. By addressing the challenges facing nursing and working collaboratively to overcome them, we can ensure that nurses are empowered to provide the highest quality care to patients and communities around the world [27]. 

In this study, we aimed to analyze currently published studies on the factors influencing nursing professionalism, assess their methodological quality, and provide recommendations for future research based on the synthesis of the best available evidence. The primary research questions guiding this study were as follows: what are the key factors influencing nursing professionalism, and how can these insights be used to enhance professional practices in nursing?

## 2. Materials and Methods

### 2.1. SALSA Method for Systematic Review

The SALSA (Search, Appraisal, Synthesis, and Analysis) method was integral to guiding our systematic review process. This method provided a structured framework that allowed us to effectively navigate the key phases of our study. Specifically, the following:Search: We defined a comprehensive search strategy to identify relevant sources of evidence on nursing professionalism.Appraisal: We critically assessed the quality and relevance of the identified studies using established evaluation criteria.Synthesis: We integrated and summarized data from the selected studies to draw clear and meaningful conclusions about the factors influencing nursing professionalism.Analysis: We interpreted the synthesized results in relation to our research question, ensuring that the findings were directly applicable to the study’s objectives.

By applying the SALSA method, we were able to manage the complexity of the systematic review process and ensure that our study was both thorough and methodologically sound.

This systematic review was conducted in accordance with the PRISMA guidelines to ensure a comprehensive and transparent reporting process.

### 2.2. Search Strategy

From 1 January 2014 to 31 January 2024, a systematic search was conducted to find studies on nursing professionalism and, in particular, to determine what factors influence professionalism. This research was conducted in the electronic databases Scopus, PubMed, Web of Science, and CINAHL. The research strategy included a combination of the five elements indicated by Terwee et al. [28]. The structure of the research is professionalism, the research population is nurses, and the exclusion filter are the main components. According to Terwee’s criteria, the exclusion filter mainly restricted publication types and topic groups. This study established that nursing competence forms the basis of comprehensive professionalism [29]. Since no measure of a single element of professionalism was considered, an assessment of professionalism was included. This review included measures of professionalism as an overall construct or as an aspect of competency [30]. In addition to the search in the electronic database, empirical secondary research was conducted by screening references and citations of the complete texts and previously published revisions.

### 2.3. Study Selection

Two researchers (VL and BD) independently assessed the suitability of the full texts using the following inclusion criteria:(1)The observed population consisted of nurses, and their specialties were based on the MeSH terms for “nursing” and “professionalism”(2)An original article, a full text in English, and journal articles that have been evaluated by others were included. Consensus was reached through discussion to resolve differences on inclusion criteria. A third auditor (IN) was responsible for the final decision if there was no consensus.
The search strings used for each individual database were as follows:**PubMed**: (Professionalism [Title/Abstract]) AND (Nurs* [Title/Abstract])*Filters*: Full text, Review, Systematic Review, published in the last 10 years, languages: Albanian, English.**CINAHL**: TI (professionalism) AND TI (nurs*);*Limiters*: Publication date from 01 January 2014 to 31 December 2024;*Expanders*: Apply equivalent subjects;*Narrow by Language*: English;*Search mode*: Find all search terms.**Web of Science (WOS)**: Professionalism (Title) AND nurs* (Title)**Scopus**: (TITLE-ABS-KEY (professionalism) AND TITLE-ABS-KEY (nurs*)) AND PUBYEAR > 2013 AND (LIMIT-TO (DOCTYPE, “re”)) AND (LIMIT-TO (LANGUAGE, “English”)).

### 2.4. Selected Research

Were identified 421 publications and 97 duplicates were eliminated. During a quick review of the titles and abstracts of the included articles, we identified and included 14 other relevant publications using the “links to similar articles” browser in the database of the researched platforms. The PRISMA (Preferred Reporting Items for Systematic Reviews and Meta-Analyses) protocol, which provides a structured approach to conducting and reporting systematic reviews, was used to present the results of the literature review [31] in Figure 1. This protocol ensures transparency and completeness in reporting the process of identifying, selecting, and synthesizing relevant studies.

### 2.5. Evaluation of Methodological Quality of the Included Studies

The evaluation of study quality is a crucial aspect in any systematic review, as it contributes to ensuring the reliability and validity of the results. In this context, the Critical Appraisal Skills Programme Qualitative Checklist (CASP) was employed, a well-established and widely recognized method for the critical appraisal of qualitative studies [32]. The choice of this approach was motivated by several fundamental reasons. Firstly, CASP has been endorsed by the Cochrane Qualitative Research Methods Group, an authoritative organization in the field of qualitative research [33]. Its approval by such a group lends CASP further credibility and establishes it as a reliable tool for assessing study quality. This is particularly important in a context where the accuracy and reliability of results are crucial for informing clinical recommendations and decisions [34]. Moreover, CASP is distinguished by its systematic approach to assessing study quality, focusing on three main areas: study validity, methodological quality, and external validity assessment [35]. This well-defined structure allows for a comprehensive and detailed examination of various aspects of research, thus providing a thorough and holistic assessment of its overall quality. The assessment of study validity, for example, focuses on examining whether the study’s conclusions are supported by the evidence presented and whether the results are reliable and accurate [36]. This entails a critical analysis of the methodologies used, potential sources of bias, and data collection and analysis strategies. Methodological quality assessment, on the other hand, focuses on evaluating the robustness of the study design and the presence of any limitations that may affect the validity of the results [37]. Lastly, external validity assessment considers the generalizability of the studies and their relevance to the population of interest, assessing whether the results can be applied to other situations or contexts [38]. Each critical appraisal was completed by the principal investigator of the systematic review work. This approach ensures consistent and accurate evaluation of the quality of the studies included in the review, minimizing the risk of bias or discrepancies in data interpretation. Additionally, it allows for a detailed and in-depth assessment of each study, thus contributing to providing a comprehensive and well-informed overview of the available evidence. In summary, the use of CASP as a tool for assessing study quality in systematic reviews offers numerous advantages, including endorsement by the Cochrane Qualitative Research Methods Group, its systematic and structured approach to quality assessment, and its intrinsic reliability and validity as an appraisal tool [39]. Through this approach, it is possible to ensure the quality and reliability of review results, thus providing a solid foundation for evidence-based clinical recommendations and decisions.

## 3. Results

### 3.1. Search

The systematic search conducted across Scopus, PubMed, Web of Science, and CINAHL databases yielded a total of 421 articles relevant to nursing professionalism. After reviewing titles and abstracts and removing duplicates, 72 studies were selected for detailed assessment.

### 3.2. Appraisal

Upon thorough evaluation, 56 out of the 72 selected studies were excluded. The primary reasons for exclusion included the lack of testing on the measurement properties of the instruments used, and relevance, as two documents focused on areas outside of nursing professionalism (one related to dentistry and the other to a general medical field). Ultimately, 14 studies met the inclusion criteria and were included in the systematic review.

### 3.3. Synthesis

The 14 included studies explored a broad range of factors influencing nursing professionalism, which can be categorized into the following key areas:

Organizational Support and Work Environment:Organizational Support: Several studies, including the one by [40]. emphasized the importance of organizational support in shaping nursing professionalism. Factors such as organizational culture and workload were identified as crucial in influencing professional behavior and attitudes.Work Environment: The research by Lombarts et al. [41] highlighted the role of standardized assessment tools and consistent work environments in maintaining high levels of professionalism among nurses.
Competency and Assessment Tools:Communication Skills and Professionalism: Pereira and Puggina [42] validated a self-assessment tool aimed at improving communication skills and professionalism among nurses, underlining the necessity of reliable and valid instruments for professional development.Measurement of Professionalism: Li et al. [43] systematic review and Vaz De Braganca and Nirmala’s [44] development of a Nurse Professionalism Scale both stressed the importance of robust measurement tools to ensure accurate assessment of professionalism in nursing.
Resilience and Well-being during the COVID-19 Pandemic:Impact of Resilience: Studies by Park and Jeong [45], as well as Zandian et al. [46] examined the impact of resilience on nursing professionalism during the COVID-19 pandemic. Their findings demonstrated how resilience mitigated job stress and supported professional behavior during crisis situations.Nurses’ Professional Well-being: The research underscored the mediating role of professionalism in ensuring nurses’ well-being, highlighting the need for resilience-building programs, especially during challenging times.

### 3.4. Analysis

The comprehensive analysis of these studies revealed the importance of targeted interventions to bolster nursing professionalism. The key areas identified include the promotion of a supportive work environment, the development of reliable competency assessment tools, and the implementation of resilience-building initiatives. The findings emphasize the critical role these factors play in enhancing nurses’ professional identity and practice, particularly in the face of crises like the COVID-19 pandemic.

### 3.5. Recommendations

Based on the synthesis of the included studies, several recommendations have been formulated to enhance nursing professionalism:

Development of Standardized Competency Assessment Tools: Ensuring consistent and accurate evaluation of nursing competencies.

Implementation of Resilience-Building Programs: Supporting nurses’ ability to manage stress and maintain professionalism during crises.

Promotion of Ethical Practices within Nursing Education and Training: Reinforcing the importance of ethics in professional nursing practice.

These recommendations are vital for guiding future research and policy-making in nursing, with the aim of fostering a more resilient and professionally competent nursing workforce.

## 4. Discussion

The systematic review provides an in-depth and comprehensive analysis of the multiple factors influencing the development and maintenance of nursing professionalism in diverse contexts. Through the exploration of a diversified corpus of studies, a complex and articulated framework emerges, highlighting the interconnection of various aspects contributing to the attainment of high professional standards.

One of the primary factors that emerges from this research is the crucial role of communication skills in establishing nursing professionalism. Studies conducted by Kim and Sim [47], Pereira and Puggina [42] and Vaz De Braganca and Nirmala [44] have meticulously examined the importance of communication skills in establishing effective therapeutic relationships with patients, facilitating interprofessional collaboration, and promoting a respectful and inclusive work environment. These findings underscore the need to invest in training and developing communication skills among nurses to enhance the quality of care and foster an organizational culture based on open and transparent communication.

Furthermore, resilience has emerged as a fundamental trait for nurses in coping with both professional and personal challenges, particularly during crises such as the COVID-19 pandemic. Studies by Park and Jung [48], Zandian et al. [46], and Jeong and Kim [49] provide a detailed analysis of how resilience influences nurses’ ability to maintain high professional standards despite the pressures and difficulties encountered in the workplace. These findings highlight the importance of promoting resilience among nurses through psychological support programs and stress management strategies to ensure their well-being and the continuity of care provided to patients.

One of the key findings from this review is the significant impact of workload and personal satisfaction on nursing professionalism, alongside the emerging recognition of resilience as a core professional behavior. Traditionally, professional behaviors in nursing have centered on elements like lifelong learning and working from a unique knowledge base, but resilience has now taken its place as an essential component. High workload has been consistently linked to increased stress levels and reduced job satisfaction, which in turn can negatively affect professional behavior and decision-making among nurses. Resilience, however, plays a critical role in enabling nurses to cope with these pressures, maintain professional standards, and provide continuous high-quality care even in challenging circumstances like the COVID-19 pandemic. Personal satisfaction similarly fosters resilience, motivation, and commitment to professional values. Addressing these factors through organizational support, work–life balance initiatives, and resilience-building programs is essential for enhancing nurses’ professional identity and improving patient care outcomes.

Another significant factor identified by the review is the importance of education and training in nursing professionalism. The articles by Fantahun et al. [40], Lombarts et al. [41], and Wang et al. [50] have emphasized the crucial role of high-quality educational programs in providing nurses with the skills, knowledge, and tools necessary to practice competently and professionally. These studies underscore the need to invest in educational and training infrastructure to ensure an adequately prepared nursing workforce capable of responding to emerging challenges in healthcare.

Finally, the perception of nursing professionalism has been recognized as influenced by psychological and cultural factors. The studies by Park and Jeong [45] and Xu et al. [51] have explored the role of professional identity, positive psychological capital, and other psychological factors in shaping nurses’ perceptions of their own professionalism and their work intentions. These findings highlight the importance of considering the cultural and psychological context in which nurses operate to develop effective strategies to enhance perceptions of professionalism and promote ethical, patient-centered nursing practice.

In conclusion, the literature review offers a rich and detailed perspective on the multiple factors contributing to the development of nursing professionalism. The identification and understanding of these factors are crucial for informing policies and practices aimed at promoting high-quality nursing practice, ensuring patient well-being, and professional satisfaction among nurses.

Table 1 summarizes the key studies analyzed in the systematic review, which involves a total of 16 countries and over 13,000 nurses. These studies, conducted between 2014 and 2023, provide a broad understanding of the factors influencing professionalism in nursing, with specific focus areas such as communication skills, job satisfaction, resilience, and professional identity.

## 5. Limitation

This study has several limitations that should be acknowledged. First, the inclusion of articles was limited to those published in English, which may have excluded relevant studies published in other languages. Second, the search was conducted only in four major databases (Scopus, PubMed, Web of Science, and CINAHL), potentially overlooking relevant studies indexed in other databases. Additionally, the scope of the review was confined to studies that specifically focused on nursing professionalism, which may have excluded broader research on related topics. Finally, while efforts were made to assess the methodological quality of the included studies, variations in study design and quality could have influenced the results and conclusions drawn from this review.

## 6. Conclusions

In summary, nursing professionalism is a multidimensional concept influenced by a wide range of factors, including communication skills, resilience, educational training, and psychological aspects. Recognizing the importance of these factors is essential for promoting high-quality nursing practice and ensuring the safety and well-being of patients. The identification of these factors provides a solid foundation for the development of targeted interventions aimed at enhancing nursing professionalism and supporting nurses in their crucial role within the healthcare system. By addressing these key elements, healthcare organizations can cultivate a culture of excellence and professionalism among nurses, ultimately leading to improved patient outcomes and enhanced healthcare delivery.

Finally, this review provides actionable recommendations derived from the synthesis of the best evidence available. These recommendations emphasize the need for targeted interventions, such as the development of reliable assessment tools and resilience-building initiatives, to foster nursing professionalism. By implementing these strategies, healthcare organizations can better support nurses in their professional development, ultimately improving patient care outcomes.

## Figures and Tables

**Figure 1 healthcare-13-00668-f001:**
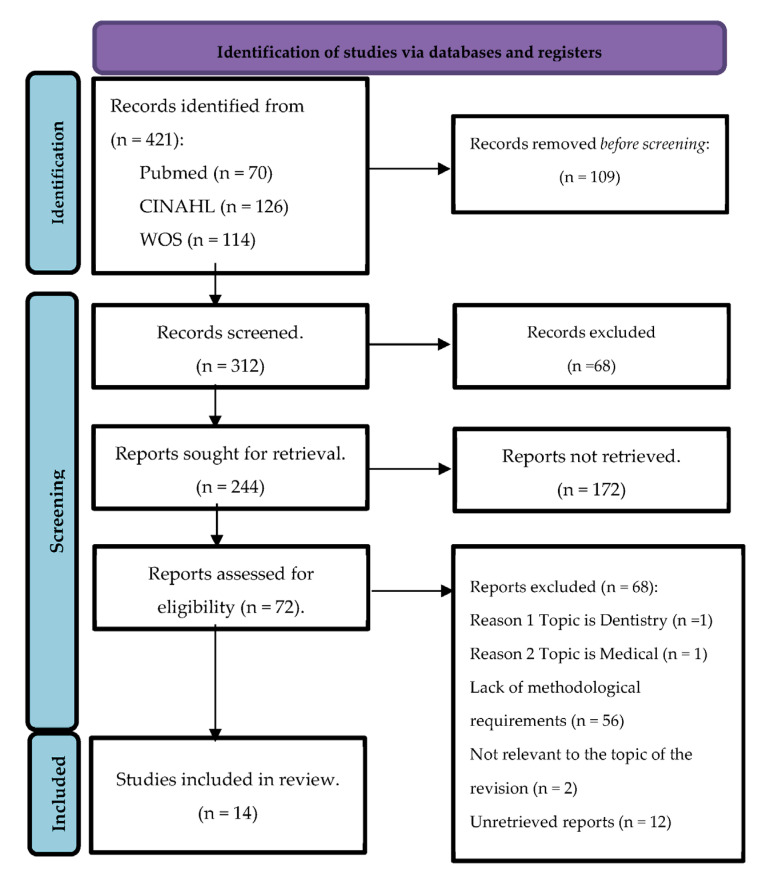
PRISMA 2020 flow diagram for systematic reviews, which included searches of databases and registers only.

**Table 1 healthcare-13-00668-t001:** Data extraction table for studies included in systematic review.

Authors	Title	Year of Publication	Study Design	Sample Size	Summary of Results
Fantahun et al. [40]	A cross-sectional study on factors influencing professionalism in nursing among nurses in Mekelle Public Hospitals, North Ethiopia, 2012	2014	Cross-sectional study	210 nurses	Found that factors influencing professionalism in nursing among nurses in Mekelle Public Hospitals in North Ethiopia include organizational culture, workload, and support systems.
Lombarts et al. [41]	Measuring Professionalism in Medicine and Nursing: Results of a European Survey	2014	Cross-sectional multilevel study	5920 Physicians and Nurses	Identified varying levels of professionalism among physicians and nurses across Europe, highlighting differences in attitudes towards professional values and behaviors.
Pereira and Puggina [42]	Validation of the self-assessment of communication skills and professionalism for nurses	2017	Cross-sectional study	110 Nurses	Validated a self-assessment tool for communication skills and professionalism among nurses, demonstrating its reliability and validity in assessing these competencies.
Li et al. [43]	Assessing medical professionalism: A systematic review of instruments and their measurement properties	2017	Systematic review	80 study inclused	Conducted a systematic review of instruments assessing medical professionalism, emphasizing the importance of valid and reliable measurement tools in evaluating professionalism.
Vaz De Braganca and Nirmala [44]	Nurse Professionalism Scale: Development and Psychometric Evaluation	2020	Cross-sectional study	1054 Nurses	Developed and evaluated the Nurse Professionalism Scale, assessing various dimensions of professionalism among nurses and demonstrating its psychometric properties.
Park and Jeong [45]	Impact of Nursing Professionalism on Perception of Patient Privacy Protection in Nursing Students: Mediating Effect of Nursing Informatics Competency	2021	Cross-sectional study	242 Nursing Students	Investigated the impact of nursing professionalism on nursing students’ perception of patient privacy protection, mediated by nursing informatics competency.
Zandian et al. [46]	Nursing work intention, stress, and professionalism in response to the COVID-19 outbreak in Iran: A cross-sectional study	2021	Cross-sectional study	362 Nurses	Examined the relationship between nursing work intention, stress, and professionalism among nurses in response to the COVID-19 outbreak in Iran, highlighting the importance of professionalism in crisis situations.
Kim and Sim [47]	Communication Skills, Problem-Solving Ability, Understanding of Patients’ Conditions, and Nurse’s Perception of Professionalism among Clinical Nurses: A Structural Equation Model Analysis	2020	Structural model analysis	171 nurses	Utilized structural equation modeling to analyze the relationship between communication skills, problem-solving ability, understanding of patients’ conditions, and nurses’ perception of professionalism among clinical nurses.
Park and Jeong [48]	Effects of the Resilience of Nurses in Long-Term Care Hospitals during on Job Stress COVID-19 Pandemic: Mediating Effects of Nursing Professionalism	2021	Cross-sectional study	200 Nurses	Explored the mediating effects of nursing professionalism on the relationship between nurses’ resilience and job stress during the COVID-19 pandemic in long-term care hospitals.
Jeong and Kim [49]	Factors influencing nurses’ intention to care for patients with COVID-19: Focusing on positive psychological capital and nursing professionalism	2022	Descriptive correlational	148 nurses	Explored factors influencing nurses’ intention to care for patients with COVID-19, focusing on positive psychological capital and nursing professionalism as determinants.
Wang et al. [50]	Study of Nightingale’s nursing professionalism: a survey of nurses and nursing students in China	2022	Cross-sectional study	1557 Nurses and Nursing Students	Conducted a survey among nurses and nursing students in China to study Nightingale’s nursing professionalism, identifying factors contributing to professional practice and development.
Xu et al. [51]	Exploring the relationship between lateral violence and nursing professionalism through the mediating effect of professional identity: A cross-sectional questionnaire study	2023	Cross-sectional study	298 Nursing Students	Explored the relationship between lateral violence and nursing professionalism among nursing students, revealing the mediating effect of professional identity in mitigating the impact of lateral violence on professionalism.
Lee and Jang [52]	Effect of Nurses’ Professionalism, Work Environment, and Communication with Health Professionals on Patient Safety Culture (AHRQ 2.0.): A Cross-Sectional Multicenter Study	2023	Cross-sectional multicenter study	271 nurses in three hospitals	Investigated the effect of nurses’ professionalism, work environment, and communication with health professionals on patient safety culture across multiple hospitals, emphasizing the importance of professionalism in ensuring patient safety.
Lee et al. [53]	Moderating Role of Communication Competence in the Association between Professionalism and Job Satisfaction in Korean Millennial and Generation Z Nurses: A Cross-Sectional Study	2023	Cross-sectional	188 nurses	Explored the moderating role of communication competence in the association between professionalism and job satisfaction among Korean millennial and Generation Z nurses, highlighting the significance of effective communication in enhancing job satisfaction.

## Data Availability

The data presented in this study are available on request from the corresponding author.

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
