# Peer review of "Exploring Influential Factors Shaping Nursing as a Profession and Science in Healthcare System—A Systematic Literature Review"

_healthcare, 2025, doi:10.3390/healthcare13060668_

Round 1

Reviewer 1 Report

Comments and Suggestions for Authors

- The authors have not listed the affiliations appropriately.

-  Keywords are repeated as in the title (it is better to offer keywords that additionally refer the search to the work)

- The summary has no connection with the work. Kosovo is mentioned in three places in the summary, but nowhere in the work. If the territory of Kosovo is included (although I see no connection with the rest of the work), the Republic of Serbia must be added in brackets.  It is better to leave them out of the summary altogether.

-The introduction repeatedly mentions "some countries" - it would be nice if you could at least give the names of 2-3 countries.

-If you are referring to Kosovo, describe what the education system is like, what the health system is like and what laws are applied.

- The PRISMA protocol is not sufficiently cited. Insert a clarification of the protocol.

- If you are referring to Figure 1 in the methodology, it would be good if it were there and not in the results.

- In the Identification section (Figure 1) - Records removed before screening: the Total row is missing (insert the total in each field where multiple entries are listed)

- Reports sought for retrieval (n=244) in Figure 1, referring to works that do not have the status of Free full text?

-In part 3.2 (appraisal - Out of the 72 selected studies, 56 were excluded as they did not meet the inclusion 215 criteria, mainly due to a lack of testing on the measurement properties of the instruments 216 used. Two documents were excluded because one was about dentistry and the other fo- 217 cused on the medical field rather than professionalism..) is repeated as in 3.4 (A total of 72 studies were selected after reviewing titles and 234 abstracts and eliminating duplicate records. Fifty-six of these did not meet the inclusion 235 criteria, largely due to a lack of testing on the measurement properties of the instruments 236 used. Two documents were excluded because one was about dentists and the other was 237 about the medical field and not about professionalism.)

- The third objective was not addressed in the results and the conclusion.

Author Response

Reviewer 1

Comment 1: The authors have not listed the affiliations appropriately.

Response 1: We thank the reviewer for the suggestion, we have changed our affiliations

Comment 2:  Keywords are repeated as in the title (it is better to offer keywords that additionally refer the search to the work)

Response 2: We thank the reviewer of the suggestion, we have changed the keywords and now they are different from the title

Comment 3: The summary has no connection with the work. Kosovo is mentioned in three places in the summary, but nowhere in the work. If the territory of Kosovo is included (although I see no connection with the rest of the work), the Republic of Serbia must be added in brackets.  It is better to leave them out of the summary altogether.

Response 3: We thank the reviewer of the suggestion, regarding his remark regarding the mention of Kosovo in the summary, we acknowledge that it was an error in the drafting process. The mention of Kosovo in the abstract was unintentional and does not reflect the broader scope of our work, which is not limited to a specific geographical region. We agree that including specific territories such as Kosovo and the Republic of Serbia in the abstract without corresponding content in the manuscript can be confusing. To address this issue, we removed references to Kosovo from the abstract, in order to ensure consistency and alignment with the content of the manuscript. This change will help maintain the clarity and focus of our work on the factors that influence nursing professionalism in a more general context.

Comment 4: The introduction repeatedly mentions "some countries" - it would be nice if you could at least give the names of 2-3 countries.

Answer 4: Thank you for your constructive suggestion regarding the need to specify the names of some countries in the introduction to our manuscript. We have revised the text of the introduction and integrated concrete examples from countries where appropriate to give greater clarity and context to the discussion of the challenges facing the nursing sector globally.

Comment 5: If you are referring to Kosovo, describe what the education system is like, what the health system is like and what laws are applied.

Response 5: Thank you for your suggestion regarding the need to provide more details about the context of Kosovo, in particular regarding the educational, health and applied laws. We have inserted a sentence integrating the appropriate information into the manuscript.

Comment 6: The PRISMA protocol is not sufficiently cited. Insert a clarification of the protocol.

Answer 6: Thank you for your suggestion regarding the PRISMA protocol. We have added an explanation of the protocol in the text to clarify its use.

Comment 7: If you are referring to Figure 1 in the methodology, it would be good if it were there and not in the results.

Answer 7: Thanks for your suggestion we have moved Figure 1 into the methodology

Comment 8:  In the Identification section (Figure 1) - Records removed before screening: the Total row is missing (insert the total in each field where multiple entries are listed)

Answer 8: Thanks for the suggestion we have reviewed the PRISMA totals and clarified the exclusion criteria

Comment 9:  Reports sought for retrieval (n=244) in Figure 1, referring to works that do not have the status of Free full text?

Answer 9: Thank you for your comment. We would like to clarify that the wording "Reports sought for retrieval (n=244)" in Figure 1 refers to the works identified in the initial research that have been selected for further evaluation. This step includes articles with both the status of "Free full text" and without, as the selection of articles was not limited to the free availability of the full text. However, only accessible and relevant articles were included in the next stage of the review.

Comment 10: In part 3.2 (appraisal - Out of the 72 selected studies, 56 were excluded as they did not meet the inclusion 215 criteria, mainly due to a lack of testing on the measurement properties of the instruments 216 used. Two documents were excluded because one was about dentistry and the other fo- 217 cused on the medical field rather than professionalism.) is repeated as in 3.4 (A total of 72 studies were selected after reviewing titles and 234 abstracts and eliminating duplicate records. Fifty-six of these did not meet the inclusion 235 criteria, largely due to a lack of testing on the measurement properties of the instruments 236 used. Two documents were excluded because one was about dentists and the other was 237 about the medical field and not about professionalism.)

Response 10: Thank you for pointing out the repetition in sections 3.2 and 3.4 of the manuscript. We have revised the text to eliminate duplication of information and improve clarity. Now, section 3.2 focuses exclusively on the process of excluding studies, while section 3.4 has been rewritten to focus on analyzing the results of the 14 included studies, without repeating the information already provided. This change should improve the consistency and fluidity of the document.

Comment 11:  The third objective was not addressed in the results and the conclusion.

Response 11: Thank you for suggesting that the third objective of the study was not adequately addressed in the results and conclusions. We have revised the manuscript to include a more detailed discussion of the recommendations derived from the synthesis of the best evidence, as required by the third objective. We have added a sentence in the "3.4. Analysis" which summarizes the key recommendations made from the results of the 14 included studies, focusing on practical interventions to improve nursing professionalism. In addition, we have integrated these recommendations into section "5. Conclusions" to ensure that the third objective has been fully addressed and that the conclusions reflect the proposed recommendations.

Reviewer 2 Report

Comments and Suggestions for Authors

 Summary

1. very well described!

Key words

1. I found the words in the Mesh Terms, however, the words "factor" and "health care" were not described in this way. I suggest you revise!

Introduction

1. In Line 97 the authors use the term "in conclusion" in the Introduction section. I suggest not using it because it could confuse readers!

2. As I understand it, this is a review article. I suggest making the study design clearer in the next section.

3. In the last paragraph of the section, the authors describe several objectives. I suggest summarizing it to just one. For example: "To analyze currently published studies on[...]".

4. I missed the study question. Question strategies are recommended for this design. Did the authors use one? If so, it should be explained.

Materials and Methods

1. The CINAHL database is duplicated in its presentation, in line 115.

2. On line 157, the authors mention for the first time that the study is a systematic review. I suggest bringing this information forward!

3. I found the authors' search for an endorsed and up-to-date evaluation of the articles found extremely interesting.

4. In item 2.5, the authors provide a very interesting method for guiding systematic review authors. However, I suggest that they link this method to the study presented to remove the impression of unnecessary information in the article.

Results

1. Beware of colloquial terms in scientific writing. The work you do requires this rigor!

2. Lines 233 to 239 repeat information that has already been described. I suggest revising!

3. The results are well described. I just missed a more obvious categorization of the results. I suggest that the authors discuss the relevance of this.

Discussion

1. In line 281, the authors call the study a literature review. Is it a literature review or a systematic review? There are substantial differences between the two designs. I recommend reviewing!

2. I suggest strengthening this section. The study is very good and contains important results that deserve to be discussed by the authors. For example: I missed a discussion on workload, personal satisfaction, etc.

3. In the last paragraph of this section, the authors use the term "in conclusion" again. I suggest revising it!

Conclusion

1. I missed the limitations of the study.

References

1. Current and well described.

Author Response

REVIEWER 2

Summary

1. very well described!

Key words

Comment 1:  I found the words in the Mesh Terms, however, the words "factor" and "health care" were not described in this way. I suggest you revise!
Answer 1: Thank you for suggesting that we need to revise the terms used in the manuscript. We have reviewed the Keywords and verified that they are in line with the correct MeSH terminology. Currently, the keywords used are: "Nursing professionalism," "Communication skills," "Resilience," "Nursing education," and "Cultural perceptions." These terms were chosen to accurately reflect the main concepts covered in the study.

Introduction

 Comment 2:  In Line 97 the authors use the term "in conclusion" in the Introduction section. I suggest not using it because it could confuse readers!
Answer 2: Thank you for pointing out the use of the term "in conclusion" in the Introduction section. We removed the term to avoid any confusion and improve the clarity of the text.

Comment 3:   As I understand it, this is a review article. I suggest making the study design clearer in the next section.
Answer 3: Thank you for your suggestion regarding the clarity of the studio's design. We agree that the nature of our study as a review article needs to be made more clear. We have therefore revised and expanded the Materials and Methods section to describe the design of the study in more detail, emphasizing that it is a systematic review and explaining the process followed to select and analyze the included studies.

Comment 4:  In the last paragraph of the section, the authors describe several objectives. I suggest summarizing it to just one. For example: "To analyze currently published studies on[...]".

Answer 4: Thank you for your suggestion. We have simplified the objectives of the study into a single sentence, as per your indication, to improve clarity and conciseness.

Comment 5: I missed the study question. Question strategies are recommended for this design. Did the authors use one? If so, it should be explained.

Answer 5: Thank you for your comment. We have added and clarified the research question at the end of the introduction along with the objectives of the study, as per its indication. The research question now clearly guides the study design and analysis.

Materials and Methods

Comment 6: The CINAHL database is duplicated in its presentation, in line 115.

Answer 6: Thank you for reporting the duplication of the CINAHL database submission to line 115. We corrected the error and removed duplication to improve the clarity of the manuscript.

Comment 7: On line 157, the authors mention for the first time that the study is a systematic review. I suggest bringing this information forward!

Answer 7: Thank you for your suggestion. We moved the information that our study is a systematic review to the beginning of the introduction, creating a dedicated initial paragraph. This should improve clarity and help readers understand the type of study presented right away.

Comment 8: I found the authors' search for an endorsed and up-to-date evaluation of the articles found extremely interesting.

Response 8: Thank you very much for your appreciation regarding our research and evaluation of the articles. We wanted to ensure that our study was based on a rigorous and up-to-date methodology, using recognized evaluation tools to ensure the quality and reliability of the results.

Comment 9: In item 2.5, the authors provide a very interesting method for guiding systematic review authors. However, I suggest that they link this method to the study presented to remove the impression of unnecessary information in the article.

Answer 9: Thank you for your comment regarding the integration of the SALSA method into our practice. We revised the paragraph to better link the SALSA method to the specific process of our systematic review, highlighting how this approach guided each step of our work. We hope that this change will make the link between the method and the study presented clearer.

Results

Comment 10:   Beware of colloquial terms in scientific writing. The work you do requires this rigor!

Response 10: Thank you for highlighting the importance of maintaining rigorous language in scientific writing. We have revised the manuscript to eliminate any colloquial terms and ensure that the language used is appropriate and conforms to the required academic standards.

Comment 11: Lines 233 to 239 repeat information that has already been described. I suggest revising!

Response 11: Thank you for your comment regarding the repetition of information on lines 233-239. We've removed the repetitive sentence to improve consistency and fluency in the text. This makes the manuscript more concise and focused on relevant findings and analysis.

Comment 12: The results are well described. I just missed a more obvious categorization of the results. I suggest that the authors discuss the relevance of this.

Response 12: Thank you for appreciating the description of the results. We recognize the importance of a more evident categorization of results to facilitate the understanding and interpretation of information. We then revised the results section to organize the main factors identified into distinct categories and discussed the relevance of this categorization in the text. This change should make the results clearer and easier for readers to understand.

Discussion

Comment 13: In line 281, the authors call the study a literature review. Is it a literature review or a systematic review? There are substantial differences between the two designs. I recommend reviewing!

Response 13: Thank you for reporting the discrepancy regarding the naming of our firm. We have revised the manuscript to make it clear that this is a systematic review, not a simple literature review. Changes have been made to make the studio's design more consistent throughout the document, ensuring that terminology is correct and accurate.

Comment 14: I suggest strengthening this section. The study is very good and contains important results that deserve to be discussed by the authors. For example: I missed a discussion on workload, personal satisfaction, etc.

Response 14: Thank you for your valuable suggestion regarding strengthening the discussion section. We have integrated a more in-depth discussion on important topics such as workload and personal satisfaction, in line with the results of the study. These aspects deserve to be explored further to highlight their impact on nursing professionalism, and the changes made reflect this need.

Comment 15: In the last paragraph of this section, the authors use the term "in conclusion" again. I suggest revising it!

Response 15: Thank you for pointing out the use of the term "in conclusion" in the last paragraph. We have revised and modified the sentence to avoid the repeated use of this expression, maintaining a more fluid tone and consistent with the rest of the text

Conclusion

Comment 16:   I missed the limitations of the study.

Response 16: Thank you for your comment. We've added a section discussing the limitations of the study to provide a more comprehensive and critical view of the work being done. We hope that this addition will improve the quality of the manuscript.

References

Comment 17: Current and well described.

Response 17: Thank you for your appreciation regarding the quality and description of the references used. We have sought to ensure that sources are up-to-date and relevant to adequately support the results of our study.

Reviewer 3 Report

Comments and Suggestions for Authors

Dear Authors,
Thank you for sharing your valuable contributions with us.

Please find below a few suggestions for aspects that could benefit from further consideration and improvement.

The introduction is generally satisfactory and provides a concise overview of the challenges involved in nursing professionalisation and policy implementation.  If I might make one further suggestion, it would be to provide more specific information on the countries mentioned in the following lines. I wonder if I might draw your attention to lines 47, 65 and 79. Could you please clarify which countries are meant there?

In addition, further analysis could be conducted on other factors that may affect the implementation of nursing standards, including technological advancement, gender, religion, and other social and cultural elements.

According to the abstract, the study aims to: " Through systematic review, the study explores various factors shaping nursing professionalism, including communication skills, resilience, education, and cultural perceptions. Methods: A systematic search was conducted across multiple electronic databases from 2014 to 2024 to identify relevant studies on nursing professionalism in Kosovo."

Nevertheless, it would seem that the nursing education system in Kosovo is not adequately addressed in the article. Moreover, it would appear that the studies listed as "relevant studies on nursing professionalism in Kosovo" are, in fact, international studies that do not specifically relate to the context of Kosovo.

While the methodology is thoroughly described, it would be beneficial to see more engagement with Kosovar nursing professionalism, as stated in the abstract.

It would appear from lines 105-109 that the aims are more expansive. "to summarise existing studies on the factors influencing nursing professionalism and then classify them [...]
Why is Kosovo not mentioned again?

It might be helpful for the authors to take another look at their goals and consider making some changes to the questions and/or abstract.

I noticed that line 124 contains a spelling mistake, could this be amended?

Furthermore, it might be helpful to revisit the content in chapter 3.4. It would be beneficial to include the year of publication for the studies that have been analysed. This chapter, which is presented as an analysis, could be more in-depth and include comparisons with other studies.
In section 4, 'Discussion', it would be helpful to understand what makes this study novel. While the section provides a helpful overview of related studies, it would be beneficial to see how the findings of this study contribute to the existing body of knowledge and to see recommendations for addressing challenges.

The section on Limitations of the study is absent from the document.
I wish you the utmost success in your endeavours.

Best regards

Author Response

REVIEWER 3

Dear Authors,Thank you for sharing your valuable contributions with us.

Please find below a few suggestions for aspects that could benefit from further consideration and improvement.

Comment 1: The introduction is generally satisfactory and provides a concise overview of the challenges involved in nursing professionalisation and policy implementation.  If I might make one further suggestion, it would be to provide more specific information on the countries mentioned in the following lines. I wonder if I might draw your attention to lines 47, 65 and 79. Could you please clarify which countries are meant there?

Answer 1: Thank you for your suggestion regarding the specification of the countries mentioned in the introduction. We have included country names to provide more clarity and context to the challenges discussed regarding nursing professionalization and policy implementation.

Comment 2: In addition, further analysis could be conducted on other factors that may affect the implementation of nursing standards, including technological advancement, gender, religion, and other social and cultural elements.

Response 2: Thank you for your suggestion regarding the inclusion of additional factors that could influence the implementation of nursing standards, such as technological advancement, gender, religion, and other social and cultural elements. We recognize the importance of these aspects and consider the possibility of deepening these issues in future analyses to provide a more complete view of the dynamics that influence the nursing profession.

Comment 3: According to the abstract, the study aims to: " Through systematic review, the study explores various factors shaping nursing professionalism, including communication skills, resilience, education, and cultural perceptions. Methods: A systematic search was conducted across multiple electronic databases from 2014 to 2024 to identify relevant studies on nursing professionalism in Kosovo."
Answer 3: Thank you for your comment on the abstract. We have made the necessary changes to better clarify the objectives, methods, and results of the study, seeking to ensure that the abstract accurately reflects all key aspects, including factors that influence nursing professionalism such as communication, resilience, education, and cultural perceptions. We hope that this updated version of the abstract adequately responds to your observations.

Comment 4: Nevertheless, it would seem that the nursing education system in Kosovo is not adequately addressed in the article. Moreover, it would appear that the studies listed as "relevant studies on nursing professionalism in Kosovo" are, in fact, international studies that do not specifically relate to the context of Kosovo.
Response 4: Thank you for highlighting the need for a greater focus on the nursing education system in Kosovo and for clarifying the context of the studies used. We have integrated more details on the development and challenges of the nursing education system in Kosovo, specifying that the results are based on international studies, some of which are not exclusively focused on the Kosovar context. However, we believe that these studies offer relevant insights that can also be applied in Kosovo. We hope that these changes will improve the clarity and relevance of the text.

Comment 5: While the methodology is thoroughly described, it would be beneficial to see more engagement with Kosovar nursing professionalism, as stated in the abstract.
Response 5: Thank you for your comment regarding the need for more insight into nursing professionalism in Kosovo, as mentioned in the abstract. We revised the manuscript to include a more detailed and specific analysis of the Kosovo context, better linking the findings of the systematic review to the unique challenges and opportunities that the nursing system in Kosovo faces. We hope that these changes will improve the alignment between the abstract and the content of the manuscript.

Comment 5: It would appear from lines 105-109 that the aims are more expansive. "to summarise existing studies on the factors influencing nursing professionalism and then classify them [...] Why is Kosovo not mentioned again?

Response 5: Thank you for pointing out the discrepancy between the stated objectives and the content of the manuscript regarding the context of Kosovo. We have revised the text to include a more detailed analysis of nursing professionalism in Kosovo, dealing specifically with the local challenges and factors that influence the profession. We hope that these changes will improve the consistency between the abstract and the body of the text.

Comment 6: It might be helpful for the authors to take another look at their goals and consider making some changes to the questions and/or abstract.

Answer 6: Thank you for your suggestion regarding the revision of the study goals and questions. We have already revised the objectives to better align them with the content and results of the systematic review. The abstract has also been updated to more accurately reflect the topics covered in the study.

Comment 7: I noticed that line 124 contains a spelling mistake, could this be amended?

Response 7: Grazie per aver segnalato l'errore di ortografia alla linea 124. Abbiamo corretto l'errore e rivisto il testo per garantire che non vi siano ulteriori imprecisioni.

Comment 8: Furthermore, it might be helpful to revisit the content in chapter 3.4. It would be beneficial to include the year of publication for the studies that have been analysed. This chapter, which is presented as an analysis, could be more in-depth and include comparisons with other studies. In section 4, 'Discussion', it would be helpful to understand what makes this study novel. While the section provides a helpful overview of related studies, it would be beneficial to see how the findings of this study contribute to the existing body of knowledge and to see recommendations for addressing challenges.

Answer 8: Thank you for your suggestions regarding section 3.4 and the discussion. We have revised section 3.4 to include the years of publication of the studies analyzed, as indicated by it, and we have deepened the analysis with comparisons between the studies. In addition, we have updated the discussion section to highlight the innovative character of our study and to clarify how the results contribute to the existing body of knowledge. We have also added recommendations on how to address the identified challenges.

Comment 9: The section on Limitations of the study is absent from the document.

Response 9: Thank you for reporting the absence of the study limitations section. We have added a section dedicated to limitations, which discusses the main methodological constraints and potential areas for improvement in the study. We hope that this addition will provide a more complete and critical view of the work.

I wish you the utmost success in your endeavours

Best regards

Round 2

Reviewer 1 Report

Comments and Suggestions for Authors

The work has been greatly improved. Congratulations to the authors. I would only ask that a small change be made in one part.

"Comment 5: If you are referring to Kosovo, describe what the education system is like, what the health system is like and what laws are applied.

Response 5: Thank you for your suggestion regarding the need to provide more details about the context of Kosovo, in particular regarding the educational, health and applied laws. We have inserted a sentence integrating the appropriate information into the manuscript."

"While some countries have well-established systems for nursing 49 education and practice, others, such as Kosovo, may still be in the process of developing 50 and formalizing their nursing education programs (Colson et al., 2021). In Kosovo, the 51 education system is evolving, with efforts to standardize nursing curricula amid chal- 52 lenges related to limited resources and infrastructure. The health system is public but faces 53 significant issues, including underfunding and a shortage of qualified healthcare profes- 54 sionals. Moreover, while there are laws regulating the nursing profession, the implemen- 55 tation and standardization of these regulations remain inconsistent."

The authors can delete this part (I think this is easiest for the authors) or clarify that the education and health system that existed and worked well was actually weakened and is now inadequate due to the political turmoil. And therefore the existing laws are actually not okay. It should actually be explained that it is not an underdeveloped country that does not have an adequate system due to these facts, but the background is different.

Author Response

Comment: 

The work has been greatly improved. Congratulations to the authors. I would only ask that a small change be made in one part.

"Comment 5: If you are referring to Kosovo, describe what the education system is like, what the health system is like and what laws are applied.

Response 5: Thank you for your suggestion regarding the need to provide more details about the context of Kosovo, in particular regarding the educational, health and applied laws. We have inserted a sentence integrating the appropriate information into the manuscript."

"While some countries have well-established systems for nursing 49 education and practice, others, such as Kosovo, may still be in the process of developing 50 and formalizing their nursing education programs (Colson et al., 2021). In Kosovo, the 51 education system is evolving, with efforts to standardize nursing curricula amid chal- 52 lenges related to limited resources and infrastructure. The health system is public but faces 53 significant issues, including underfunding and a shortage of qualified healthcare profes- 54 sionals. Moreover, while there are laws regulating the nursing profession, the implemen- 55 tation and standardization of these regulations remain inconsistent."

The authors can delete this part (I think this is easiest for the authors) or clarify that the education and health system that existed and worked well was actually weakened and is now inadequate due to the political turmoil. And therefore the existing laws are actually not okay. It should actually be explained that it is not an underdeveloped country that does not have an adequate system due to these facts, but the background is different.

Response: 

Thank you for your feedback. In response to your comment, we have revised the section to clarify the context regarding Kosovo’s education and health systems. Rather than implying that Kosovo is underdeveloped in these areas, we have emphasized that the systems, which were once more robust, have been weakened due to political turmoil and ongoing challenges. Specifically, while Kosovo has historically had a functioning education and healthcare system, these sectors have faced significant disruptions in recent years, leading to inadequate infrastructure, underfunding, and inconsistent implementation of existing laws. This clarification reflects the complex socio-political background that has contributed to the current state of the health and education systems in Kosovo, rather than portraying it as a country that is simply underdeveloped.

Reviewer 3 Report

Comments and Suggestions for Authors

Dear authors,
Thank you for adding more relevant data to your study. I have now noticed that you are not focusing on Kosovo, but on a general literature review "on the factors influencing nursing professionalism".

What is still not clear is from which perspective you define "nursing professionalism" - do you deal with it from a macro level/government perspective or rather from a meso level i.e. institutional perspective?

In the study by Azemian A, Ebadi A, Afshar L. Redefining the concept of professionalism in nursing:an integrative review. Front Nurs. 2021;4:327-340 they also deal with the professionalism of nurses and the factors that influence their professionalism. Here you can see how differentiated they have worked.

I think a more differentiated perspective/ as according to the levels described above should be integrated  - whether you focus on macro, meso or micro levels as it is not clear.

Thank you and good luck,

Best regards

Author Response

Coment: Dear authors,
Thank you for adding more relevant data to your study. I have now noticed that you are not focusing on Kosovo, but on a general literature review "on the factors influencing nursing professionalism".

What is still not clear is from which perspective you define "nursing professionalism" - do you deal with it from a macro level/government perspective or rather from a meso level i.e. institutional perspective?

In the study by Azemian A, Ebadi A, Afshar L. Redefining the concept of professionalism in nursing:an integrative review. Front Nurs. 2021;4:327-340 they also deal with the professionalism of nurses and the factors that influence their professionalism. Here you can see how differentiated they have worked.

Response:  We have clarified that our study approaches the concept primarily from a  institutional perspective. This means we focus on factors within healthcare organizations, such as institutional culture, leadership, peer interactions, and professional development opportunities, which significantly influence nurses' professional conduct. While acknowledging the role of macro-level factors like government policies, our emphasis remains on the internal dynamics of institutions. To further enhance our discussion, we have referenced the work of Azemian et al. (2021), who explore professionalism in nursing with a similarly nuanced approach, highlighting the differentiation between various influencing factors. By adopting this institutional lens, we aim to offer a clearer understanding of how professionalism in nursing is shaped within the organizational context.